# A Mixed Thermosensitive Hydrogel System for Sustained Delivery of Tacrolimus for Immunosuppressive Therapy

**DOI:** 10.3390/pharmaceutics11080413

**Published:** 2019-08-14

**Authors:** Hsiu-Chao Lin, Madonna Rica Anggelia, Chih-Chi Cheng, Kuan-Lin Ku, Hui-Yun Cheng, Chih-Jen Wen, Aline Yen Ling Wang, Cheng-Hung Lin, I-Ming Chu

**Affiliations:** 1Department of Chemical Engineering, National Tsing Hua University, Hsinchu 300, Taiwan; 2Center for Vascularized Composite Allotransplantation, Department of Plastic and Reconstructive Surgery, Chang Gung Memorial Hospital, Chang Gung Medical College and Chang Gung University, Taoyuan 333, Taiwan; 3Graduate Institute of Biomedical Sciences, College of Medicine, Chang Gung University, Taoyuan 333, Taiwan

**Keywords:** allotransplantation, hydrogels, sustained delivery, tacrolimus

## Abstract

Tacrolimus is an immunosuppressive agent for acute rejection after allotransplantation. However, the low aqueous solubility of tacrolimus poses difficulties in formulating an injection dosage. Polypeptide thermosensitive hydrogels can maintain a sustained release depot to deliver tacrolimus. The copolymers, which consist of poloxamer and poly(l-alanine) with l-lysine segments at both ends (P–Lys–Ala–PLX), are able to carry tacrolimus in an in situ gelled form with acceptable biocompatibility, biodegradability, and low gelling concentrations from 3 to 7 wt %. By adding Pluronic F-127 to formulate a mixed hydrogel system, the drug release rate can be adjusted to maintain suitable drug levels in animals with transplants. Under this formulation, the in vitro release of tacrolimus was stable for more than 100 days, while in vivo release of tacrolimus in mouse model showed that rejection from skin allotransplantation was prevented for at least three weeks with one single administration. Using these mixed hydrogel systems for sustaining delivery of tacrolimus demonstrates advancement in immunosuppressive therapy.

## 1. Introduction

Hydrogels are composed of crosslinked polymeric networks that can form highly hydrated semisolid materials. Because their diverse capacity to carry therapeutic agents in a depot-like form allows sustained release, hydrogels have been utilized in many applications, including drug delivery, cell therapy, and tissue engineering [1,2,3,4].

Among various types of hydrogels, thermosensitive poly(ethylene glycol)-*co*-poly(amino acids) hydrogels are capable of in situ gelling upon injection to appropriate sites in the body. They can deliver drugs or cells to specific sites with minimum invasiveness while serving as a depot for extended delivery [5,6,7,8,9]. These polymers belong to LCST-type materials; thus, when above a certain temperature, their aqueous solution undergoes a phase transition to the gel state, depending on the concentration. The gelling process involves a reduced hydrogen bond interaction between the hydrophilic segments of polymers and water as the temperature rises [10,11]. Different types of hydrophilic polymers are used in this gelling process, including various hydrogels, poly(ethylene glycol) (PEG) [12], methoxy poly(ethylene glycol) (mPEG) [13], and PEO–PPO–PEO copolymers (mainly the Pluronic^®^ or poloxamer (PLX) series) [14]. Several types of polypeptides have been used in the hydrophobic portion of the polymer [12,15,16]. The fact that peptides have secondary interactions between or within themselves complicates the gelling process of these amino acid-based hydrogels, creating different types of self-assembled morphology [17,18,19,20]. Additionally, because peptide bonds possess low hydrolytic rates, compared with ester bonds, for example, hydrogels degrade at extremely slow rates. Therefore, side effects caused by long-term foreign body reactions may be a concern, or the release rates of some drugs might be too slow for intended applications.

In this study, a suitable formulation of a hydrogel carrier was sought for tacrolimus, an immunosuppressive, antirejection drug. Tacrolimus is a hydrophobic substance with low aqueous solubility. Controlled delivery of tacrolimus would avoid the low bioavailability problem and improve patient compliance statistics by reducing the administration frequency. Several delivery systems for tacrolimus have been reported, including mPEG-poly(lactic acid) nanoparticles [21], poly(lactide-*co*-glycolide) microspheres [22], and triglycerol monostearate hydrogels [23]. The in situ gelation property of the system studied here has several advantages over particulate systems or conventional hydrogel systems, including longer release time, higher encapsulation efficiency, easy administration, and applicability to various types of drugs. In particular, by combining two types of hydrogel components, the possibility of increasing the flexibility of carriers to suit distinct pharmaceutical entities and modifying release rates for better clinical outcomes is explored.

## 2. Materials and Methods

### 2.1. Materials

*O*,*O*’-Bis(2-aminopropyl) polypropylene glycol-block-polyethylene glycol-block-polypropylene glycol (Poloxamer, PLX, PPG–PEG–PPG, Mn = 900), l-alanine, hydrogen bromide (HBr/acetic acid), and a LIVE/DEAD staining kit were purchased from Sigma-Aldrich (St. Louis, MO, USA). Tacrolimus (FK506) was obtained from LC Labs (Woburn, MA, USA). Trifluoroacetic acid-d (TFA-d) and trifluoroacetic acid (TFA) were purchased from Alfa Aesar (Wall Hill, MA, USA). l-lysine-(Z) was commercially obtained from ACROS (Morris Plains, NJ, USA). Diethyl ether, hexane, and ethanol (95%) were purchased from Echo Chemicals (Toufen, Miaoli, Taiwan). Dimethyl sulfoxide (DMSO) was obtained from J.T. Baker. Dulbecco’s modified eagle’s medium (DMEM), fetal bovine serum (FBS), and antimycotics–antibiotics were purchased from Gibco (Grand Island, NY, USA). Toluene and tetrahydrofuran (THF) were obtained from TEDIA (Fairfield, OH, USA), and chloroform, dichloromethane (DCM), and *N*,*N*-dimethylformaide (DMF) were purchased from AVANTOR (Center Valley, PA, USA). All solvents were dried over CaH_2_ before used.

### 2.2. Synthesis of l-alanine N-Carboxyanhydride (Ala-NCA), l-Lysine-(Z) N-Carboxyanhydride (Lys-(Z)-NCA), and P–Lys–Ala–PLX Block Copolymer

The copolymer synthesis was similar to that in our previous study conducted in 2016 [14] in which l-alanine (5.0 g and 56.1 mmol) and triphosgene (16.65 g, 56.3 mmol) were dissolved in anhydrous THF (150 mL) and gently stirred at 45 °C under nitrogen flux. After the reaction period of approximately 12 h, the mixture turned clear and the solution was condensed to a final volume of 15 mL through rotary evaporation. The product was precipitated in excess *n*-hexane. A similar preparation method was employed in this study: l-lysine-(Z) (5.0 g, 17.8 mmol) using 3.2 g of triphosgene (10.7 mmol) was dissolved in 60 mL of anhydrous THF. The reaction was completed within 4 h, and the mixture was then condensed and finally precipitated.

The block copolymer P–Lys–Ala–PLX was prepared through a two-step ring-opening polymerization of Ala–NCA and Lys–(Z)–NCA with PLX as the macroinitiator. Briefly, PLX (4.0 g, 5 mmol) was dissolved in 15 mL of toluene and azeotropically distilled to a final volume of approximately 5 mL to remove residual water. Ala-NCA (5.198 g) was added to the reaction flask and dissolved in 20-fold of anhydrous chloroform/DMF (2:1). The mixture was stirred at 40 °C for 24 h until the reaction was complete and then precipitated with diethyl ether. Next, pre-weighed Lys–(Z)–NCA (3.063 g) was added into the reaction flask, and the reaction was carried out at 40 °C for 24 h again. The final product was precipitated with diethyl ether.

The copolymer (Z)–Lys–Ala–PLX–Ala–Lys–(Z) was dissolved in TFA (1 g/10 mL), and a solution of 33 wt % HBr/acetic acid (0.3 mL/mol l-lysine) was added and stirred at 0 °C for 1 h. After precipitation with diethyl ether, the product was dialyzed (MWCO 1000) using a spectrum dialysis bag, lyophilized, and stored in a vacuum atmosphere for future use.

### 2.3. ^1^H Nuclear Magnetic Resonanc (NMR) Spectroscopy

A ^1^H NMR spectrum of the 0.5 wt % copolymer solution in TFA-d was obtained using a Varian UNITY INOVA 500 MHz spectrometer (Palo Alto, CA, USA) to verify the chemical structure of the product.

### 2.4. Fourier-Transformed Infrared Spectroscopy (FT-IR)

Fourier-transformed infrared spectroscopy (FT-IR) was performed using an FTIR spectrometer (Nicolet™ iS50, Thermo Fisher Scientific, Waltham, MA, USA) equipped with an attenuated total reflectance (ATR) module. The copolymer solution prepared at 5 wt % in deionized water was injected into the test chamber. Spectra were collected at a temperature ranging from 10 to 50 °C at increments of 10 °C. The sample was equilibrated for 20 min at each temperature. The amide I band region of 1600–1700 cm^−1^ was used to collect information on secondary structures and conducted for chemical structure verification.

### 2.5. Sol-to-gel Phase Transition

The sol-to-gel transition behavior of copolymers prepared at various concentrations was investigated using the test tube inversion method. Samples were prepared in test tubes at 3 to 7 wt % in deionized water and placed in a tube rotator overnight at 4 °C until the solutes were completely dissolved. The inverted test tube test was performed at a temperature range of 10–50 °C, at increments of 1 °C. At each temperature, the sample was allowed to equilibrate for 10 min. The gelation point was designated when the solution stopped flowing while inverted and gently agitated.

### 2.6. Scanning Electron Microscope (SEM)

The copolymer mixed gel solution was prepared at 5 wt % in P–Lys–Ala–PLX and 1 wt % in Pluronic F-127 that allowed full equilibration at 4 °C overnight for future use. The hydrogel microstructure was examined using a JEOL JSM-7001F SEM system (Peabody, MA, USA). Briefly, hydrogels were gelled at 37 °C and then submerged in liquid nitrogen prior to lyophilization. To investigate the cross section, lyophilized hydrogels were carefully cut open using a scalpel blade.

### 2.7. Degradation Test

The in vitro degradation of hydrogels (100 μL) in phosphate-buffered saline (PBS) and PBS containing elastase (5 U/mL) was analyzed. Briefly, copolymer solutions were gelled in 1.5 mL tubes at 37 °C for 30 min prior to adding 1 mL of the respective degradation mediums. The medium was replaced at each time point, and the residual was weighted after washing and lyophilization.

### 2.8. Biocompatibility

The cell compatibility of the prepared mixed hydrogel was assessed. Briefly, human embryonic kidney cells (293T) were loaded onto a 24-well transwell plate at 6 × 10^3^ cells per well and grown in DMEM media, supplemented with 10% FBS and 1% antimycotics–antibiotics. Exactly 100 μL of the copolymer solution was gelled at 37 °C in the transwell insert overlaid with 1 mL of culture media. LIVE/DEAD staining with fluorescence microscopy observation was performed on the third and seventh days to quantitate the viability of the cells, and a 3-(4,5-dimethylthiazol-2-yl)-2,5-diphenyltetrazolium bromide (MTT) assay was simultaneously performed to detect cell proliferation. Briefly, cells were treated with 3-(4,5-dimethylthiazol-2-yl)-2,5-diphenyltetrazolium bromide (10% *v*/*v* in medium) in each well for 3 hours before 200 μL of DMSO was added. The solution was collected and analyzed.

### 2.9. Drug Release In Vitro

The hydrogels P–Lys–Ala–PLX 5 wt % and Pluronic F-127 1 wt % were combined, solubilized in deionized water, and then mixed with tacrolimus in tubes to afford 1 mL solutions of the final drug concentration with 10 or 20 mg/mL. A negative control of the mixed hydrogel without tacrolimus was used. 100 μL copolymer solutions were loaded into a 1.5 mL tube and kept at 37 °C for 10 min before adding 1 mL of PBS with 2% tween 20 as the conditional medium. At each time point, aliquots were collected using HPLC analysis to ascertain the tacrolimus concentration. A linear calibration curve over the concentration range of 1000 μg/mL to 1 μg/mL was constructed, and the tacrolimus concentration was interpolated accordingly.

### 2.10. Drug Release In Vivo

Male Lewis (LEW) rats, weighing 300 to 350 g, were obtained from the National Laboratory Animal Center and were housed under pathogen-free conditions at the Animal Center of Linkou Chang Gung Memorial Hospital according to protocols approved by Institutional Animal Care and Use Committee of Chang Gung Memorial Hospital (IACUC No. 2017121807, approved date, 14 May 2018). Those rats (*n* = 3) were injected subcutaneously with 1 mL mixed hydrogels loaded with tacrolimus at 10 mg/mL. To analyze the in vivo release rate of tacrolimus, whole blood was drawn from the tail vein of those rats on days 7, 14, and 28 and put into the VACUETTE^®^ EDTA tubes. Whole blood concentration of tacrolimus was quantitated by liquid chromatography-tandem mass spectrometry (LC-MS/MS) (Waters, Milford, MA, USA) at the Department of Laboratory Medicine, Linkou Chang Gung Memorial Hospital. The performance characteristics and assay method has been described previously. [24] The method for tacrolimus assay showed linearity over the calibrator range, *r* > 0.999, with the range measurement: 1.1–38.5 ng/mL and accuracy 108%. For intra-day and inter-day precision, the CV was <6.6%. This method showed no carryover or ion suppressant. The calibration solution (6PLUS1^®^ RMultilevel Calibrator Immunosuppressants in Whole Blood) was purchased from Chromsystems (Munich, Germany) and contained tacrolimus and a blank calibration fluid (blank calibrator)). Further, 100 μL of calibration solution, quality control solution, and sample were loaded into a 1.5 mL centrifuge tube, and 300 μL of 0.1 M zinc sulfate containing internal standards (FK-506) were added. The mixtures were then vigorously vortexed for 30 s then centrifuged at 12,000 rpm for 15 min to completely precipitate the protein. The 300 μL supernatant was removed, placed in a 96-well collection tray, and put into the autosampler.

## 3. Results and Discussion

### 3.1. The Synthesis and Characterization of P–Lys–Ala–PLX and Pluronic F-127

Thermosensitive hydrogels composed of amphiphilic block copolymers were used in this study, specifically PLX-poly(l-alanine-lysine) (P–Lys–Ala–PLX). The ^1^H-NMR spectra of the triblock copolymer are illustrated in Figure 1A, where peaks corresponding to copolymers are observed and successful synthesis of copolymers is confirmed. The molecular weights, as measured by Gel Permeation Chromatography (GPC) and ^1^H NMR, are displayed in Table 1. Briefly, the molar mass polydispersity of the triblock copolymer is 1.51, and the weight average molecular weight (Mw) is 3817 Da.

ATR-FTIR was used to verify the synthesis of the copolymer P–Lys–Ala–PLX and compared with that of *O*,*O*’-Bis(2-aminopropyl) polypropylene glycol-block-polyethylene glycol-block-polypropylene glycol (Poloxamer, PLX), which has the same characteristic peaks as the PLX segment of P–Lys–Ala–PLX. As in Figure 1B, peaks at 3300 cm^−1^ (N–H bond) and 1600–1720 cm^−1^ (amide bond) are from polypeptides, while those found at 2900 cm^−1^ (C–H) and 1100 cm^−1^(C–O) are from the PLX segment. The characterization and structure of the triblock copolymer P–Lys–Ala–PLX is explained in our 2016 study [14].

The sol–gel transition phase diagram is represented in Figure 2. In the room to body temperature range, P–Lys–Ala–PLX underwent a sol-to-gel transition in the concentration range from 3% to 7%. The gelation temperature decreased as the concentration of copolymers increased. For solutions with a copolymer concentration lower than 3 wt %, hydrogels were unable to form stable gels under the inverted test tube test despite significant increases in viscosity at higher temperatures. Consequently, copolymer concentrations of 4 to 5 wt % were chosen for further formulation studies.

The internal structure of the P–Lys–Ala–PLX hydrogel was examined through scanning electronic microscopy, as portrayed in Figure 3. A plate-like structure was dominant with some fibrous features within. This type of morphology is typical for PLX-poly(alanine) hydrogels.

### 3.2. Development of Mixed Hydrogel Formulation

After the preliminary in vivo drug release tests with the P–Lys–Ala–PLX hydrogel, extremely slow tacrolimus release rates and a considerably low plasma concentration of tacrolimus were observed. Unexpected transplant rejection resulted from these tests (data not shown) because drug concentrations were low. The slow in vivo release rate of the drug may be caused by the slow degradation rate of the hydrogel in vivo. Therefore, the hydrogel formulation should be modified to accelerate drug release at effective concentrations and sustain the release for more than 30 days. Pluronic F-127 has been mentioned to modulate the drug release rate [25]. The addition of Pluronic F-127, an approved fast-degrading hydrogel, to the P–Lys–Ala–PLX hydrogel system may provide a solution to accelerate the release rate.

A series of hydrogel formulations of 1 to 3 wt % Pluronic F-127 powder mixed into 4 or 5 wt % P–Lys–Ala–PLX were studied, as shown in Table 2. Sol-to-gel transition properties, cytotoxicity, drug encapsulation efficiency, and the release rate were measured. The formulation of 5 wt % P–Lys–Ala–PLX with 1 wt % Pluronic F-127 (sample 5:1) had a lower transition temperature than the other three groups, as demonstrated in Figure 4A. Additionally, the sample 5:1 had the highest drug encapsulation efficiency (Figure 4B). Consequently, the formulation of the sample 5:1 was chosen for further studies.

### 3.3. Biodegradability

Degradation behavior of hydrogels in a buffer solution with and without the enzyme elastase at 37 °C was observed. P–Lys–Ala–PLX gels (5 wt %) incubated with elastase (5 U/mL) resulted in a mass loss of 40% after 14 days, as displayed in Figure 5, whereas gels incubated with the buffer solution lost only 10% of the original mass. The sample 5:1 incubated with elastase had a mass loss of 50% after 14 days, whereas those without the enzyme lost only 15% of the original mass. The mass loss of mixed hydrogels was higher than that of the P–Lys–Ala–PLX hydrogel because Pluronic F-127 was more prone to dissolve back to an aqueous solution. In short, hydrogels were stable when proteinases were absent, and Pluronic F-127 accelerated degradation rates.

SEM micrographs of lyophilized P–Lys–Ala–PLX and a sample 5:1 mixed hydrogel are shown in Figure 6. The mixed hydrogels exhibited more porous structures and space resulting from the incorporation of Pluronic F-127. After the 14-day degradation, samples of mixed hydrogels revealed larger pores and looser structure resulting from the faster dissolution or disintegration of Pluronic F-127.

### 3.4. Biocompatibility

The dispersion of insoluble tacrolimus particles in the hydrogel solution was rather challenging [21,22]. Certain solvents must be used to dissolve tacrolimus during the encapsulation process to assure an even distribution of the drug. Acceptable solvents should be nontoxic to cells and not interfere in the sol–gel transition of the polymeric system.

Several solvents that can dissolve tacrolimus were chosen, including DMSO, ethanol, and transcutol P. Ethanol had the least toxicity and highest dissolving power when water was present (data not shown). To reduce the toxicity of ethanol to the cells, tacrolimus was dissolved in ethanol as 100 mg/mL or 200 mg/mL stock solutions, which were then used to prepare the 10 mg/mL and 20 mg/mL drug-carrying hydrogels, respectively. Tacrolimus encapsulated in the gel gradually precipitated out as small evenly dispersed solid particles when ethanol diffused into the surrounding aqueous environment. When the gel pellet was placed in a 10-time volume of medium and the residual ethanol concentration was below 1 wt %, its conditioned medium exerted little effect on the cells in the in vitro toxicity test (see Figure 7).

Cellular compatibility under encapsulation of prepared hydrogels was evaluated using LIVE/DEAD staining, in which live cells stained green and dead cells were red. To evaluate the cytotoxicity of materials, in the medium, cells were cultivated in contact with hydrogels in transwells. The cells showed acceptable viability in culture for 3 to 7 days with both P–Lys–Ala–PLX and mixed hydrogels, as related in Figure 8.

### 3.5. Tacrolimus Encapsulation and Release

The feasibility of hydrogels as an efficient local drug delivery system for tacrolimus was studied first under in vitro conditions. The encapsulation efficiency and release of tacrolimus in mixed hydrogel were measured. The values of encapsulation efficiency (EE) of the sample 5:1 mixed hydrogel were measured at two levels of drug content, where EE = (experimental drug loading)/(theoretical drug loading) × 100%. The results are presented in Table 3. The data indicated the hydrogel encapsulated tacrolimus rather well, with the EE at approximately 96%–98% in triplicate samples. Drug release rates in P–Lys–Ala–PLX or the sample 5:1 mixed hydrogel was measured for a one-month period. The release rates were stable and higher in the mixed hydrogel at both drug content levels, as illustrated in Figure 9. No burst release was observed, and the sustained release of tacrolimus was achieved. Release rates correlated with hydrogel degradation (e.g., see Figure 5) and thereby suggest that the enhanced disintegration of mixed hydrogels was responsible for higher drug release rates in mixed hydrogels. These results demonstrate the ability to modify the release rate by using mixed hydrogels.

For the mixed hydrogel with 10 mg/mL tacrolimus group, preliminary in vivo drug release tests were conducted. The aim was to establish a correlation between our in vitro release data and the corresponding data in a plasma concentration when 1 mL of the formulation was injected subcutaneously in a mouse. More comprehensive pharmacokinetic work, together with allotransplantation animal study, is currently underway, and results will be presented in a separate paper. As exhibited in Table 4, plasma concentrations of tacrolimus during a 28-day period after administration were stable at approximately 10 ng/mL, a level generally considered clinically relevant [23,26]. Therefore, the linear in vitro release rates reflected consistent in vivo plasma concentrations.

## 4. Conclusions

Tacrolimus is a potent immunosuppressive age that can be used for patients who receive allotransplantation. The current daily or twice-daily oral dosage forms may encounter patient compliance problems. Encapsulation of tacrolimus by in situ gelling hydrogels is a feasible way to deliver this drug. Using the mixed hydrogel system developed in this study, steady and extended delivery was achieved. The use of toxic solvents was avoided and the formulation exhibited high biocompatibility with 293T cells in vitro with high encapsulation efficiency of drug and good gelling properties. Preliminary in vivo data also confirm a stable plasma concentration of the drug for an extended period of time. In summary, the newly designed formulation of mixed hydrogels is a promising delivery system for tacrolimus and, perhaps, other highly hydrophobic compounds.

## Figures and Tables

**Figure 1 pharmaceutics-11-00413-f001:**
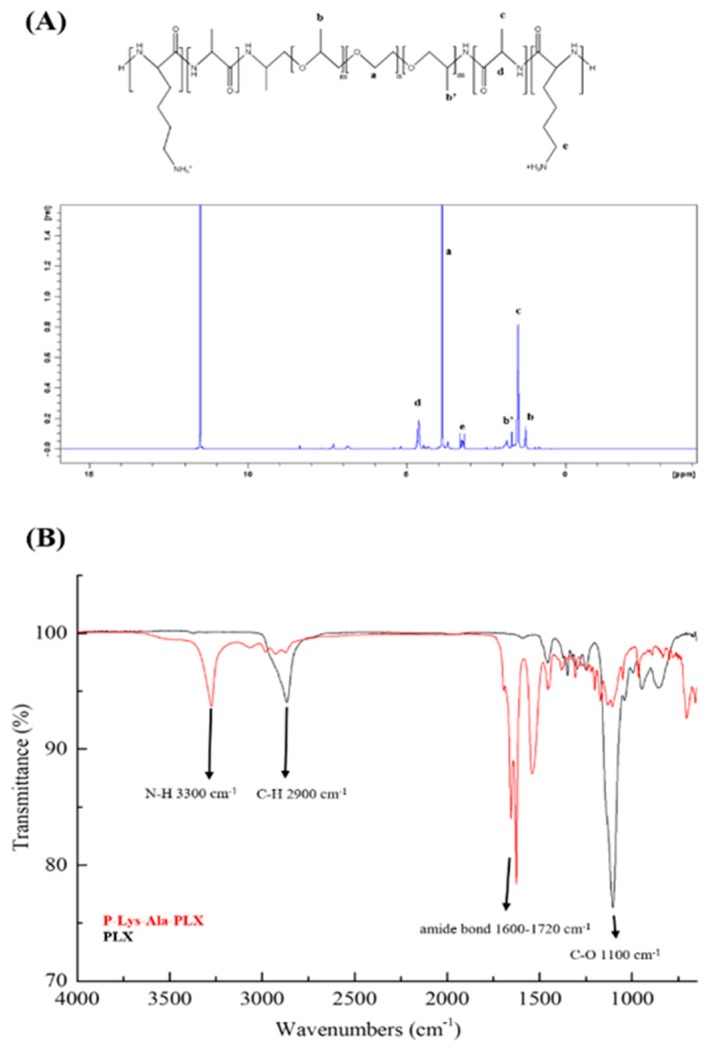
(**A**) ^1^H-NMR of poloxamer (PLX)-poly(l-alanine-lysine) (P–Lys–Ala–PLX) copolymer in trifluoroacetic acid-d (TFA-d). (**B**) FT-IR spectra of P–Lys–Ala–PLX (red line) and PLX (black line) samples obtained via an attenuated total reflectance (ATR) module (2900 cm^−1^, alkane of Pluronic F-127; 3345 cm^−1^, amide of poly(l-alanine) and poly(l-lysine); 1637 cm^−1^ and 1540 cm^−1^, amide I and II of poly(l-alanine) and poly(l-lysine); 1100 cm^−1^, carbon-oxygen bond of PLX.

**Figure 2 pharmaceutics-11-00413-f002:**
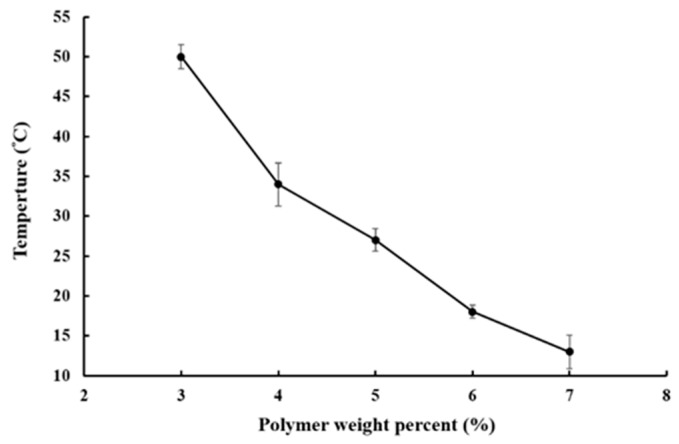
Sol–gel transition profile of aqueous solutions of P–Lys–Ala–PLX.

**Figure 3 pharmaceutics-11-00413-f003:**
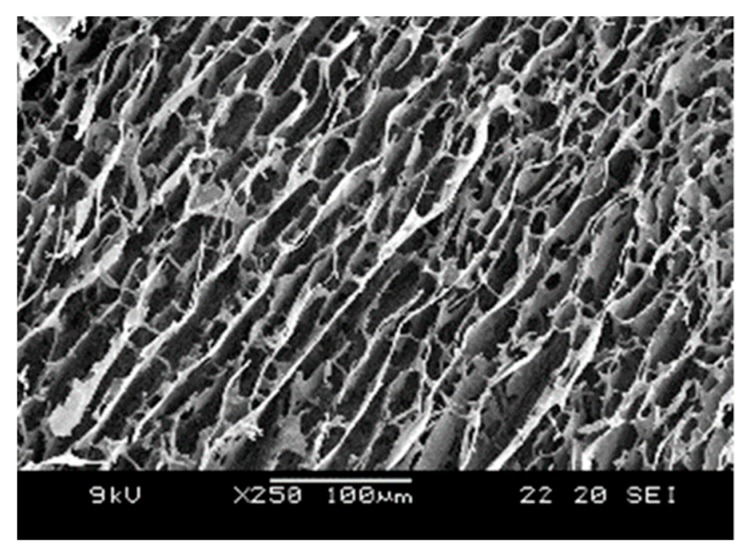
SEM image of P–Lys–Ala–PLX hydrogels.

**Figure 4 pharmaceutics-11-00413-f004:**
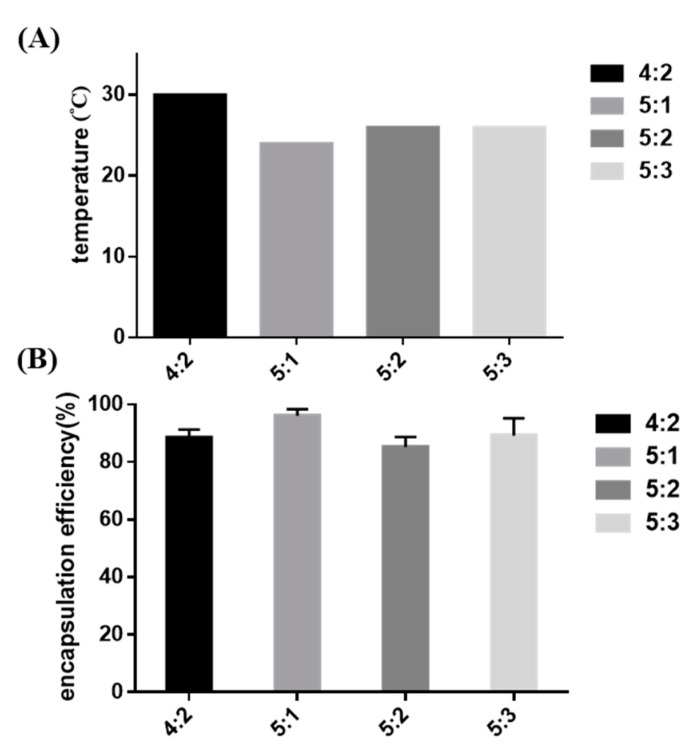
(**A**) The sol-to-gel transition temperature of mixed hydrogels without tacrolimus; (**B**) the encapsulation efficiency of tacrolimus in mixed hydrogels.

**Figure 5 pharmaceutics-11-00413-f005:**
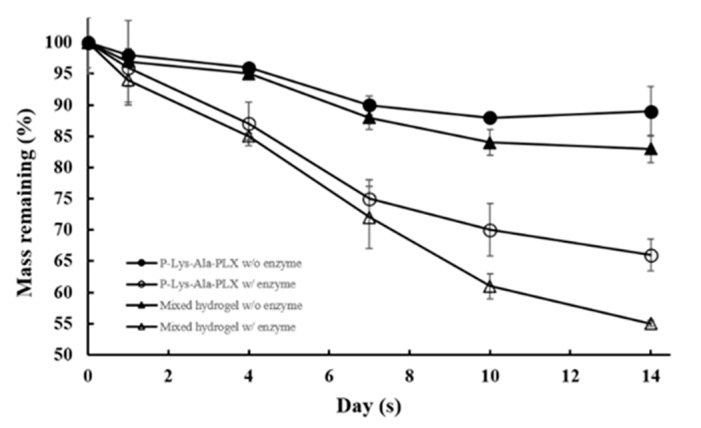
Degradation profiles of P–Lys–Ala–PLX and sample 5:1 mixed hydrogel in phosphate-buffered saline (PBS) with or without 5 U/mL elastase (*n* = 6).

**Figure 6 pharmaceutics-11-00413-f006:**
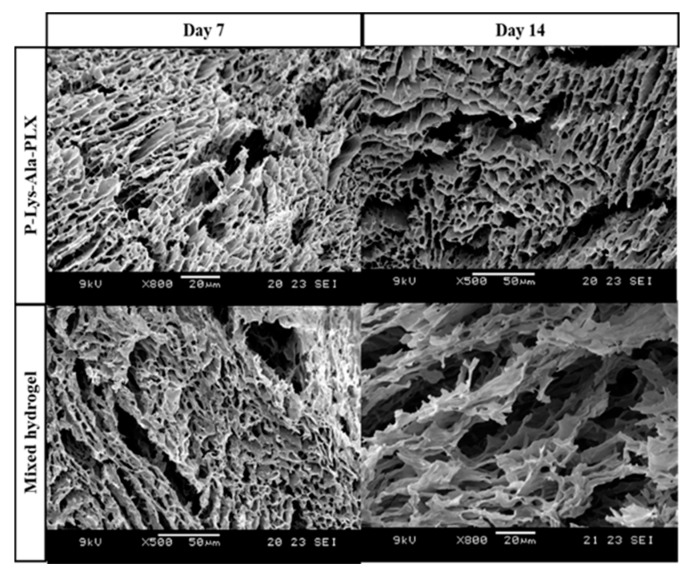
SEM microphotographs of P–Lys–Ala–PLX hydrogel and sample 5:1 mixed hydrogel after degradation in PBS with 5 U/mL elastase at 37 °C for 7 and 14 days.

**Figure 7 pharmaceutics-11-00413-f007:**
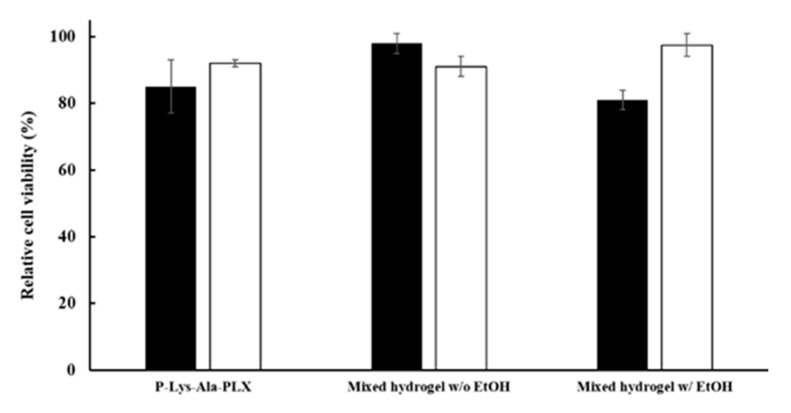
Cytotoxicity of conditioned mediums of various formulations by the MTT assay. Mixed hydrogels: 5:1 P–Lys–Ala–PLX to Pluronic F-127. Ethanol was added to the gel as in the 10 mg/mL formulation but without the drug. The black bar was on day 3 (black) and the white bar was on day 7.

**Figure 8 pharmaceutics-11-00413-f008:**
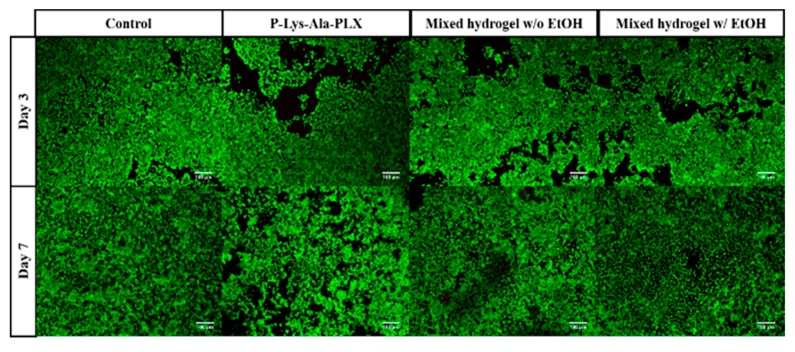
The LIVE/DEAD staining of 293T cells was co-cultured with various hydrogels without the drug.

**Figure 9 pharmaceutics-11-00413-f009:**
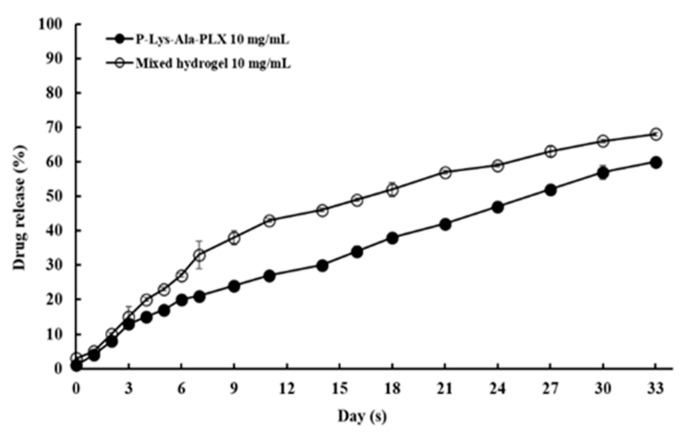
In vitro tacrolimus release as a fraction of total encapsulated drug.

**Table 1 pharmaceutics-11-00413-t001:** The molecular weight and polydispersity as determined by GPC and ^1^H NMR.

	Ala	Lys	Mn ^a^	Mw ^b^	PDI ^b^
P–Lys–Ala–PLX	19.4	1.8	2528	1.51	3817

^a^. Determined by ^1^H NMR. ^b^. Determined by GPC.

**Table 2 pharmaceutics-11-00413-t002:** The series formulation test group of the mixed hydrogels.

Group	P–Lys–Ala–PLX	Pluronic F-127
1	4 wt %	2 wt %
2	5 wt %	1 wt %
3	5 wt %	2 wt %
4	5 wt %	3 wt %

**Table 3 pharmaceutics-11-00413-t003:** The encapsulation efficiency of 5:1 mixed hydrogel for tacrolimus.

Group	Encapsulation Efficiency (%)
10 mg/mL	96.13 ± 1.56
20 mg/mL	98.5 ± 0.97

**Table 4 pharmaceutics-11-00413-t004:** Whole blood concentrations of tacrolimus: In vivo drug release results from mixed hydrogels containing 10 mg/mL tacrolimus.

**Tacrolimus (ng/mL)**	**Time (Days)**		
	7 (*n* = 3)	14 (*n* = 3)	28 (*n* = 3)
	8.5 ± 1.5	12.2 ± 2.1	10.1 ± 1.6

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
