# Peer review of "A Mixed Thermosensitive Hydrogel System for Sustained Delivery of Tacrolimus for Immunosuppressive Therapy"

_pharmaceutics, 2019, doi:10.3390/pharmaceutics11080413_

Round 1

Reviewer 1 Report

The concept of injectable sustained release preparations for tacrolimus is of interest clinically.  I would be interested to see some discussion on how dose might be adjusted in the event of under or over-exposure.  What impact would pyrexia be predicted to have on tacrolimus release from the gel?  this is a well-written paper and the data presented support the conclusions drawn.

I would like to raise several minor specific issues:

1) P2L74:  probably mean explain rather than explicated.

2) Fig 7:  need key for black and white bars.

3) Performance characteristics should be provided for the tacrolimus assay.

Author Response

What impact would pyrexia be predicted to have on tacrolimus release from the gel?
Author response: Thank you for your insightful comments. 1. The extended delivery system used here is designed as a subcutaneously implanted device. Dose adjustment of this kind of system can be made by removing the device with simple surgery with replacement another more suitable implant or by supplement of other dosage forms. The easy monitoring and constant plasma concentration from the implant can facilitate dose adjustment if needed. 2. Physiology parameters, such as local temperature may affect the release rate from the device. The variations may be small since gel properties do not change much above the gelling temperature for the hydrogels we used. The temperature variation due to pyrexia is relatively small to fundamentally alter the drug release rate.

1) P2L74: probably mean explain rather than explicated.

Reply: Thank you for the corrections. Modifications were made accordingly.

2) Fig 7: need key for black and white bars.

Reply: Thank you for the corrections. Modifications were made accordingly.

3) Performance characteristics should be provided for the tacrolimus assay.

Reply: As suggested by the reviewer, we added a new paragraph into the manuscript (section 4.10)

“Performance characteristics have been reported previously26. The method for tacrolimus assay showed linearity over the calibrator range, r>0.999, with the range measurement: 1.1-38.5 ng/mL and accuracy 108%. For intra-day and inter-day precision, the CV was <6.6 %. This method showed no carryover or ion suppressant.”

Reviewer 2 Report

The manuscript is well written. The experimental section has been adequately described. Results are original and interesting and adequately supported by the experiments.

Author Response

Reply: Thank you. We are very grateful for your comments. We have revised it in our new version as the attachment for your reference.

Reviewer 3 Report

It is a well written manuscript. The content and rationale from Chemistry and Pharmacy perspective is sound. The data on in vivo release needs further clarification. Although I understand that the in vivo data will be published later. There does need to be more details in this manuscript.

Please add details of assay method for determining tacrolimus in blood. Were the concentrations of tacrolimus in blood determined by HPLC/MS methods? What was the limit of sensitivity of assay to determine tacrolimus? I assume that the assay was blood based and not plasma based.

Please also correct the title Table 4: Whole blood concentrations of tacrolimus: .....Please also add the number on animals on which the mean data are based.

Author Response

Reply: First, we would like to thank the reviewer for your comments, which help to improve the quality of this manuscript.

About the assay method:

As suggested by the reviewer, we added the below paragraph into the manuscript (section 4.10).

“The calibration solution (6PLUS1® RMultilevel Calibrator Immunosuppressants in Whole Blood) was purchased from Chromsystems (Munich, Germany) and contained tacrolimus and a blank calibration fluid (blank calibrator)). A hundred μL of calibration solution, quality control solution and sample were loaded into a 1.5 mL centrifuge tube, and 300 μL of 0.1 M zinc sulfate-containing internal standards (FK-506) were added. The mixtures were then vigorously vortex for 30 seconds then centrifuged at 12000 rpm for 15 minutes to completely precipitate the protein. The 300 μL supernatant was removed, placed it in a 96-well collection tray and put it into the autosampler.”

About the instrument used:

Thank you for your comment. We used UPLC-MS/MS Ultra-performance Liquid Chromatography Tandem Mass Spectrometric Method for the drug release in vivo study. This information was included in the abovementioned paragraph.

About the limit of sensitivity of the assay,

Analytical measurement ranges from 1.1 - 38.5 ng/ml. This information is included in the abovementioned paragraph.

About whether the assay was blood-based or plasma-based.?

We are very sorry for the typo. Yes, we collect the whole blood for the drug release in vivo study.

Please also correct the title Table 4: Whole blood concentrations of tacrolimus: ...... Please also add the number of animals on which the mean data are based.

Thank you for the corrections. We have revised it and add the number of animals on each time point measurement. (Please see Table 4.)

One reference was added in the revision.

Huang, P.-C. L., Ya-Ching Huang, Yu-Shao Chiang, Huey-Ling You, Chia-Ni Lin, Hsiao-Chen Ning. Development of an ultra-performance liquid chromatography tandem mass spectrometric method for simultaneous quantitating four immunosuppressant drugs. J Biomed Lab Sci 2015;27.